# Urea Decomposition Mechanism by Dinuclear Nickel Complexes

**DOI:** 10.3390/molecules28041659

**Published:** 2023-02-09

**Authors:** Christian O. Martins, Leticia K. Sebastiany, Alejandro Lopez-Castillo, Rafael S. Freitas, Leandro H. Andrade, Henrique E. Toma, Caterina G. C. Marques Netto

**Affiliations:** 1Department of Chemistry, Universidade Federal de Sao Carlos, Rod. Washington Luiz, s/n, km 235, Sao Carlos 13565-905, Brazil; 2Department of Chemical Engineering, Universidade Federal de Sao Carlos, Rod. Washington Luiz, s/n, km 235, Sao Carlos 13565-905, Brazil; 3Institute of Physics, Universidade de Sao Paulo, Rua do Matao, 1371, Sao Paulo 05508-220, Brazil; 4Institute of Chemistry, Universidade de Sao Paulo, Av. Prof. Lineu Prestes 748, Sao Paulo 05508-900, Brazil

**Keywords:** nickel complexes, urea decomposition, urease mimics, ancillary ligand

## Abstract

Urease is an enzyme containing a dinuclear nickel active center responsible for the hydrolysis of urea into carbon dioxide and ammonia. Interestingly, inorganic models of urease are unable to mimic its mechanism despite their similarities to the enzyme active site. The reason behind the discrepancy in urea decomposition mechanisms between inorganic models and urease is still unknown. To evaluate this factor, we synthesized two *bis*-nickel complexes, [Ni_2_L(OAc)] (1) and [Ni_2_L(Cl)(Et_3_N)_2_] (2), based on the Trost *bis*-Pro-Phenol ligand (L) and encompassing different ligand labilities with coordination geometries similar to the active site of jack bean urease. Both mimetic complexes produced ammonia from urea, (1) and (2), were ten- and four-fold slower than urease, respectively. The presence and importance of several reaction intermediates were evaluated both experimentally and theoretically, indicating the aquo intermediate as a key intermediate, coordinating urea in an outer-sphere manner. Both complexes produced isocyanate, revealing an activated water molecule acting as a base. In addition, the reaction with different substrates indicated the biomimetic complexes were able to hydrolyze isocyanate. Thus, our results indicate that the formation of an outer-sphere complex in the urease analogues might be the reason urease performs a different mechanism.

## 1. Introduction

Urease is a nickel-containing enzyme responsible for urea hydrolysis into carbon dioxide and ammonia (Figure 1A). The active site of urease accommodates two pseudooctahedral [1,2] or 5-coordinate nickel centers bridged by a carbamylated lysine and a hydroxide. The coordination sphere of these nickel centers is completed with two histidine residues each plus a terminal water or an asparagine residue (Figure 1B,C) [3,4]. Mechanistic models of urease describe the coordination of urea to Ni 1 via its carbonyl and the involvement of the bridged hydroxide as a nucleophile in the formation of carbamic acid as an intermediate, which self-decomposes into carbon dioxide and ammonia [4].

Synthetic models of enzymes are designed based on the enzymatic active site and on its catalytic mechanism. The design of a biomimetic complex focuses on the recognition of specific interactions to obtain an adequate parallel between natural and synthetic catalysts [5]. This parallel offers a rationalization behind these unique biologic systems [6], serving as an excellent tool to study enzymatic mechanisms. Several inorganic models have been described in the literature [7,8,9,10,11,12,13,14,15] to aid the discussion of urease’s catalytic mechanism. However, ammonia has only been detected as a decomposition product in a few of these complexes despite several of these models able to bind urea in a monodentate and a bidentate mode, [10,15,16,17]. These urease analogues describe the formation of isocyanate as an intermediate. Previously, this species was considered as a possible intermediate of urease enzyme [18], even though the study of Callahan et al. confronted this hypothesis after observing the hydrolysis of a fully methylated urea [19].

The inability of urease mimics to hydrolyze urea with the same mechanism of urease is intriguing, especially since several of these models have a striking similarity towards the enzyme active site. Motivated by this, we decided to synthesize urease models and study their catalysis to explore the factors contributing to an elimination mechanism rather than hydrolysis. Here, we report a study of the influence of the ligand lability on two novel urease biomimetic complexes encompassing acetate or chloride ions as labile ligands. The *bis*-nickel complexes, (1) and (2), were based on (*RS*,*RS*)-Trost-bis-ProPhenol(L) as a ligand (Figure 1D). In relation to the enzyme active center, these model systems provide an aromatic environment around the nickel centers. The phenolate bridge mimics the hydroxide bridge between the nickel centers, and the two pyrrolidines act as two of the four histidine analogues (e.g., His529 and His409). Their behavior can be compared to the urease from jack bean, allowing exploration of the reactivity differences and improving the comprehension of the elimination mechanism of mimetic complexes. Computational analyses, ^15^N NMR assays, and catalytic experiments indicated the order of ligand substitution during the reaction using the analogues influences the resulting outcome. These results help to elucidate the mechanism of urea decomposition by urease mimetic complexes.

## 2. Results and Discussion

### 2.1. General Characteristics of (1) and (2) Complexes

Complexes (1) and (2) were synthesized from Trost (*RS*,*RS*)-Bis-ProPhenol ligand in acetonitrile in the presence of nickel acetate or nickel chloride, respectively. Consistent data were obtained from microanalysis, UV-Vis spectroscopy (Figure 2), magnetic measurements (Figure 3), infrared spectroscopy (SI1), mass spectroscopy (SI2-3), conductivity, cyclic voltammetry, and spectroelectrochemistry (SI4 and SI5). Both complexes were isolated as uncharged species and (2) exhibited a color change from yellow to greenish blue upon dissolution in acetonitrile, whereas (1) remained green. Interestingly, (1) did not undergo dissociation into ionic species in acetonitrile (40 μScm^−1^), whereas (2) formed a 1:1 electrolyte (108.5 μS cm^−1^ in the conductivity). Therefore, the color change of (2) in addition to the value of its solution conductivity indicate a fast ligand substitution between chloride and solvent, confirming the different reaction rates by changing the labile ligand.

Both complexes presented d-d transitions ascribed to an octahedral field splitting in the UV-Vis spectroscopy of their acetonitrile solutions (Figure 2). For instance, (1) exhibited d-d bands with maximum absorptions at 500 nm (ε = 3.1 dm^3^ mol^−1^ cm^−1^) and 680 nm (ε = 3.2 dm^3^ mol^−1^ cm^−1^), which could be ascribed to ^3^A_2_→^3^T_1g_ (F) and ^3^A_2_→^3^T_1g_ (P) based on comparisons to other nickel complexes of the literature. Complex (2) exhibited bands at 463 nm (ε = 9 dm^3^ mol^−1^ cm^−1^), 563 nm (ε = 6.7 dm^3^ mol^−1^ cm^−1^), 606 nm (ε = 7 dm^3^ mol^−1^ cm^−1^), 655 (ε = 6 dm^3^ mol^−1^ cm^−1^), and 688 nm (ε = 5.5 dm^3^ mol^−1^ cm^−1^). The observed shoulders at the 600–700 nm region for both complexes could be ascribed as a spin flip transition ^3^A_2_→^1^E_g_, which is consistent to other octahedral nickel complexes of the literature [20,21]. Interestingly, (2) exhibits characteristics of a symmetric complex in which both nickel centers have similar electronic features. The (2) complex seems to have two asymmetric nickel centers, as four bands are observed in the region of ^3^A_2_→^3^T_1g_ (F) and ^3^A_2_→^1^E_g_, transitions.

The nickel centers of both complexes interact magnetically with each other in the solid state. This has been demonstrated in the experiment of temperature dependence of the magnetic susceptibility in which χ=M/H and the χT product are displayed in Figure 3 for both nickel complexes. As shown in Figure 3, a characteristic paramagnetic behavior down to T = 2 K in the $\chi \left(T\right)$ curve is observed for both samples due to the absence of any indication of a magnetic phase transition. The χT product deviates from the Curie law at high temperatures (T > 50 K), showing a slight increase (more noticeable for (2)) below ~25K, followed by a fast drop with decreasing temperature. This behavior is an indication of the development of ferromagnetic interactions between nickel centers (intradimer interaction) and a subsequent antiferromagnetic coupling between dimers (interdimer interaction) at low temperatures. To describe the magnetic behavior of this binuclear system, we have used the Heisenberg exchange Hamiltonian H = −2JS_1_.S_2_, from which the following expression for the susceptibility can be derived:(1)$$\chi \left(T\right)=\left(1-P\right)\chi_{dim}\left(T\right)+2P\chi_{P}\left(T\right)+2N$$
where $\chi_{dim}\left(T\right)=\frac{N_{A}g^{2}\mu_{B}^{2}}{k\left(T-\theta \right)}\times \frac{2e^{\left(\frac{2J}{kT}\right)}+10e^{\left(\frac{6J}{kT}\right)}}{1+3e^{\left(\frac{2J}{kT}\right)}+5e^{\left(\frac{6J}{kT}\right)}}$, and $\;\chi_{P}\left(T\right)=\frac{N_{A}g^{2}\mu_{B}^{2}}{3kT}$

Here, P ≈ 10^−6^ corresponds to a small fraction of the paramagnetic impurity, and N ≈ 10^−4^ emu/Oe mol refers to the temperature-independent paramagnetism. To consider the interaction among dimers, we incorporated a phenomenological Weiss constant, θ. As shown in Figure 3, the expression (1) gives a good description of the data with the relevant parameters, g = 2.1, J = 0.67 cm^−1^, θ = −0.94 K and g = 2.0, J = 0.81 cm^−1^, θ = −0.72 K for (1) and (2), respectively. The positive values for the intradimer exchange couplings J and negative Weiss constants confirm the presence of the ferromagnetic and antiferromagnetic interactions for both nickel complexes.

To evaluate if the coupling between nickel centers remained in solution, both complexes were submitted to cyclic voltammetry coupled to spectrolectrochemistry experiments. Evidence of an electronic coupling between the nickel centers was only perceived for (1) in the cathodic scan (SI4). There are two overlapped waves at −0.36 and −0.88 V, which can only be ascribed to the reduction of the coupled Ni(II) ions, considering the lack of compatible reducing centers in the ligand. The spectral changes of increased bands centered at 266, 292, and 384 nm indicate a substantial perturbation of the electronic structure of the complex along with the reduction process. If the two waves were associated with the bridged Ni(II)/(I) centers, their separation of 0.52 V would be consistent with a comproportionation constant (K_c_) of 6 × 10^8^, reflecting a strongly coupled system in agreement with the magnetic susceptibility experiments. In the anodic scan, two undistinguishable processes were observed, which could be associated with two electrons each, as determined by DPV (SI5). The first process is centered around 0.98 V, is correlated with the increase of the bands at 269 nm, 285 nm, 354 nm, and 421 nm, and is ascribed to the oxidation of phenol moiety. This oxidation process intensified the ligand-to-metal charge-transfer bands at 354 nm and 421 nm due to a possible change in the coordination geometry. The second process at 1.24 V was associated with the decrease of both the p-p* transition and LMCT of Ni^2+^, whereas an increase of a band centered at 396 nm was observed. This band was ascribed to an oxidation of both nickel centers as the LMCT band of Ni^3+^ complexes are around 380 nm [22,23]. In the case of (2), no reduction process could be observed and only the anodic processes were evaluated by spectroelectrochemistry (SI6). Interestingly, the spectral changes were not as evident as for (1), indicating the smaller electronic coupling accompanying the substitution of the chloride-bridging ligand by solvent molecules.

### 2.2. Catalytic Behavior of the Biomimetic Complexes

Complexes (1) and (2) were shown to encompass several structural characteristics of the urease active site. For instance, by UV-Vis spectroscopy both complexes were observed to present an octahedral geometry, and the solid-state magnetic susceptibility measurements indicated both nickel centers were coupled. Interestingly, cyclic voltammetry coupled to spectroelectrochemistry indicated (1) maintained the Ni-Ni coupling in solution, whereas (2) seems to suffer ligand substitution of chloride and Et_3_N by solvent molecules, disrupting the Ni-Ni coupling. However, despite having some structural similarities to urease’s active site, the catalytic decomposition of urea should be tested. Moreover, if these complexes exhibit catalytic activity, then different behaviors are expected since (1) remained uncharged in solution with a more stable geometry, whereas (2) was charged upon dissolution and presented a lower symmetry (Figure 2).

Urea hydrolysis experiments were quantified by the amount of ammonia released over time by using Berthelot’s method [24]. Urease enzyme was used as a control, presenting an initial reaction rate of 7 × 10^−3^ μmol s^−1^ of NH_3_ at a 6.6 mM initial urea concentration. Complexes (1) and (2) were only ten-fold (0.8 × 10^−3^ μmol s^−1^ of NH_3_) and four-fold (1.75 × 10^−3^ μmols^−1^ of NH_3_) slower than urease, respectively (Figure 4). The difference in reaction rates between the chloride and acetate complexes is assigned to a dissociative mechanism from ligand substitution kinetics, controlled by the more labile chloride ligand over acetate. This feature is in agreement with the conductivity measurements since the uncharged (2) could form a 1:1 electrolyte in acetonitrile solution, whereas (1) remained uncharged in acetonitrile, indicating the rate determining step involves a ligand substitution step.

Considering ligand substitution is important for catalysis, we decided to explore in more detail the reaction mechanism of urea decomposition by (1) and (2) complexes. Seven possible intermediates were proposed to be formed during the urea hydrolysis reaction (intermediates 1a–1g, Figure 5). These intermediates differ in the order of ligand substitution and in the mode of urea interaction with the complex (path 1–3, Figure 5). For instance, intermediates 1b and 1c would form with urea coordination to the nickel centers, whereas intermediates 1e and 1g would bind urea by secondary interactions. As urea coordination to the nickel center has been observed in the urease enzyme [3], our initial hypothesis was that a similar interaction would occur in our complexes. Therefore, the possibility of forming intermediates 1b or 1c was assessed by coordinating urea to complexes (1) and (2) in acetonitrile and analyzing the products via FTIR spectroscopy. As expected for intermediates 1b and 1c, bands at 3432 and 3335 cm^−1^ for free νNH_2_, 3210 cm^−1^ for bond νNH_2_ and 1658 (δs NH_2_), 1637 (δas NH_2_), and 1614 cm^−1^ for C=O were observed (SI7) [25]. This is possible due to urea bonded in the bidentate and monodentate mode in the [Ni_2_L(H_2_O)_3_(Urea)]^+^complex [13,14]. Therefore, [Ni_2_L(Urea)]^+^ (intermediate 1b or 1c) would be a possible intermediate during urea hydrolysis. However, a stepwise reaction between (1) and urea followed by addition of stoichiometric amounts of water in anhydrous acetonitrile resulted in no detection of ammonia, suggesting urea hydrolysis in these complexes most probably proceeds via complexes 1e or 1g.

Thus, to support the hypothesis of complex 1e and/or 1g formation, water titration of a (1) solution, followed by conductivity measurements was performed. The presence of intermediates 1e and/or 1g in solution would imply the production of a 1:1 electrolyte. Beyond precipitation, an increase in conductivity from 40 μS cm^−1^ to 132 μS cm^−1^ was observed upon water titration, indicating the apparent formation of complexes 1e or 1g. Solid IR analysis of the soluble fraction (SI8) presented bands at 1698 cm^−1^ and 1477 cm^−1^ that could be from a protonated carboxylate [26] or from monodentate ν_a_ COO^−^ and ν_s_ COO^−^ (Figure 4, complex 1d) [27]. However, titration with deuterated water revealed the same spectra as H_2_O titration (SI8), indicating the soluble complex is most likely from a monodentate acetate (complex 1a). The insoluble fraction obtained with D_2_O titration presented bands at 2546 cm^−1^ from antisymmetric OD stretching and 1260 cm^−1^ from MOH stretching, which upon deuteration is shifted to 1230 cm^−1^ (SI9). These bands evidence the formation of a hydroxo complex 1f. These intermediates were shown to be essential for urea hydrolysis when stoichiometric urea added to (1) dissolved in a water/acetonitrile system resulted in the formation of ammonia. Therefore, it seems water should first coordinate to the nickel centers, forming [Ni_2_L(H_2_O)_2_]^+^ before the urea interaction. Complex 1d, obtained from the water titration experiments, enabled us to analyze the importance of this complex in the reaction since upon addition of an anhydrous urea solution to an anhydrous solution of complex 1d, stoichiometric amounts of ammonia were produced (0.23 μmol). A control reaction, using an anhydrous solution of (1) complex, resulted in no ammonia formation, strengthening the idea of an aquo/hydroxo complex acting as the active species in catalysis.

In the jack bean urease active site (PDB: 4GY7), two of the three coordinated water molecules are possibly aquo-species, whereas the bridging water is suggested to be a hydroxo-form at the optimum pH [28]. The bridging hydroxide was described to act as a nucleophile in the reaction [29,30,31,32]. Therefore, if the bis-nickel complexes behaved similar to urease, then the aquo (1d) or hydroxo (1f) intermediates should behave as a nucleophile, and intermediates 1e and/or 1g should undergo a hydrolysis generating carbamic acid. However, NMR experiments using ^15^N urea solution indicate the complexes perform a different mechanism than urease. Surprisingly, the ammonia/ammonium signal could not be detected, even when urease was the catalyst, despite the presence of bubbles in the NMR tube assumed to be ammonia. Upon (1) addition, the signal corresponding to urea (73 ppm) disappeared, and a new signal at 243 ppm could be detected, which was assigned to isocyanate (Figure 6B). Isocyanate was formed in an elimination mechanism in which an aquo or a hydroxo ligand acts as a base to deprotonate the amide nitrogen from urea [18]. Curiously, in the case of complex (2), urea was never completely consumed, even at longer reaction times since both signals (73 ppm and 243 ppm) were evident (Figure 6B). The steady condition observed for (2) complex can be expected if there is a dynamic equilibrium between reactants and products, and the addition of acid or bicarbonate should shift the equilibrium (Figure 6A). Isocyanate concentration was lowered upon acid addition, whereas in the presence of bicarbonate, the equilibrium was completely shifted towards urea (Figure 6C). The equilibrium shift is only possible if isocyanate is further hydrolyzed to carbon dioxide and ammonia; therefore, it is reasonable to imply that one water molecule acts as a nucleophile, while a second water molecule acts as a general base during isocyanate hydrolysis [33].

It should be noted that all urease mimics in the literature that form ammonia undergo the elimination mechanism [15,16,17,34]. Barrios and Lippard [15] reasoned that coordinated urea increases its acidity upon N-coordination, which facilitates the deprotonation by a nickel-bond hydroxide. In our experiments, urea decomposition does not stop at isocyanate as it can be further hydrolyzed. The elimination followed by hydrolysis mechanism indicates a hydrolytic ability of intermediates 1e and 1g in contrast to other nickel complexes mimetic of urease [16,34,35]. This feature could be due to the employed ligand since it has been described as an important tool in dictating the orientation of electrophiles and nucleophiles by coordination of the substrates within the chiral pocket [36]. This type of ternary complex resembles many enzymatic mechanisms [37]. In the case of our complexes, the presence of a hydrophobic pocket can lower the pK_a_ of the water molecule bond to nickel [38], forming a hydroxo-complex (intermediates 1d and 1g) able to deprotonate the bridged coordinated urea.

To validate that the isocyanate intermediate can undergo hydrolysis under the reaction conditions, several other compounds were tested as substrates for (1) and (2) complexes and the urease enzyme. In these catalytic reactions, ammonia formation was quantified by the Berthelot method (Table 1), and the formation of other molecules was analyzed by gas chromatography (GC-MS). Intriguingly, there was a similarity in the yields of urease [39] and both complexes when urea, N-methylurea, formamide, and butylcarbamate were used as substrates (Table 1). For acetamide, an elimination mechanism involves the formation of an unstable product, which resulted in low conversion yields for both nickel complexes (yield up to 4%). In contrast, urease could perform acetamide hydrolysis in high yields (82%). Urease was unable to hydrolyze N-phenylurea though (1) and (2) complexes could use an elimination reaction to generate phenyl isocyanate, followed by a hydrolysis reaction to form aniline (SI10). Hence, we confirmed the hydrolysis of isocyanate can take place in a bis-nickel coupled system and that urease undergoes a different mechanism from that observed for our complexes. This result is in agreement to the Barrios and Lippard study [15] in which when a 50% aqueous solution of acetonitrile was employed, isocyanate was not appreciably detected, suggesting it was being further hydrolyzed. It has been inferred that the positioning of the urea amide close to a bridging hydroxide would be the reason behind the detection of isocyanate instead of carbamate since the hydroxide can serve as a general base in an elimination mechanism [10,15,40]. This hypothesis was corroborated by theoretical models, and urea was shown to coordinate a nickel atom via its oxygen atom while forming a hydrogen bond through the NH_2_ group to the bridging OH ligand [34]. This step is followed by a proton transfer from the bridging H_2_O ligand to one of the nitrogen atoms of urea with a subsequent removal of a proton of the amide pointing towards the bridging hydroxide ligand, forming isocyanate and ammonia [34].

### 2.3. Theoretical Calculations of the Possible Mechanisms

To corroborate our hypothesis, theoretical calculations on the transition states of a self-elimination mechanism and a catalyzed elimination mechanism with [Ni_2_L(urea)]^+^ (species II, Figure 7A) complex were compared to a transition state obtained with [Ni_2_L(H_2_O)_2_]^+^ complex (species X, Figure 7A). Urea was coordinated in a bridged bidentate configuration in [Ni_2_L(Urea)]^+^ complex due to its higher stability over the monodentate coordination, which has a binding energy near 292 kJ.mol^−1^. Contrastingly, the acetate monodentate is slightly more stable than the corresponding bidentate mode. In general, the urea complex is 441 kJ.mol^−1^ less stable than the acetate, indicating a difficulty in substituting acetate by urea.

Sequential addition of two water molecules to the [Ni_2_L(Urea)]^+^ complex (Path 1, Figure 7A) resulted in dissociation (OH^−^ + H^+^) of the first water molecule (species III, Figure 7A), followed by coordination and hydrogen bond stabilization by the second water molecule (species IV, Figure 7A). Consequently, coordination of the first water molecule to one of the nickel centers led to the formation of a hydroxo-complex with the tertiary alcohol from the ligand holding the proton release. This water addition has an energetic cost near to 36.3 kJ.mol^−1^ in relation to [Ni_2_L(Urea)]^+^ complex plus one isolated water molecule (conversion of species II to III, Figure 7). Stabilization of the [Ni_2_L(Urea)]^+^-hydroxo-complex is possible with the addition of the second water molecule (species IV, Figure 7) due to its hydrogen bond polarization near the hydroxyl group. The stabilization energy of IV was 71.9 kJ.mol^−1^ relative to the first water molecule addition, resulting in a net stabilization of 35.6 kJ.mol^−1^ from the initial energy of [Ni_2_L(Urea)]^+^ (species II, Figure 7). The respective transition state from IV to V results in an energy barrier of 437 kJ mol^−1^ (Figure 7). Hence, the formation of [Ni_2_L(Urea)]^+^ complex seems to inhibit any hydrolytic reaction. Ammonia self-elimination from the [Ni_2_L(Urea)]^+^ (species II, path 2, Figure 7A) complex resulted in a transition state 221 kJ.mol^−1^ higher in energy giving an isocyanate coordinated to the [Ni_2_L] complexes as an elimination product (species IX, Figure 7). Calculations starting from the [Ni_2_L(Urea)]^+^ complex and ammonia resulted in the elimination of water to obtain coordinated isocyanate. These results are in agreement to our experimental analysis in which a [Ni_2_L(Urea)] was unable to produce ammonia, suggesting that if urea coordinates before water, then a well of energy stability is reached to suppress any hydrolytic reaction.

Interestingly, the aquo complex (1d Figure 5 and species X, Figure 7), was theoretically calculated to be 52 kJmol^−1^ more stable than the [Ni_2_L(Urea)]^+^ (species II, Figure 7) complex, as shown in Figure 7. Urea coordination by secondary interactions to the [Ni_2_L(H_2_O)_2_]^+^ complex results in a net stabilization energy of −102 kJ mol^−1^ (state XI, Figure 7). After urea interaction through the oxygen, the free amide nitrogen receives a hydrogen from the amide nitrogen bond to water to form ammonia and isocyanate (transition state 235 kJ mol^−1^ higher than [Ni_2_L(H_2_O)_2_]^+^-urea) (support information video). Moreover, the interaction between coordinated waters and urea seems to have a deep influence on the reaction transition state. If the interaction takes place through one of the nitrogen atoms, then the transition state is only 90 kJmol^−1^ (Figure 7b, XI (N) and XII (N)). Based on these results and our experimental data, urea decomposition seems to pass through the [Ni_2_L(H_2_O)_2_]^+^ complex, which stabilizes urea by secondary interactions. This forces it to undergo an elimination reaction to form isocyanate, which is further hydrolyzed into ammonia and carbon dioxide.

These results corroborate our experimental data since once urea coordinates, water cannot attack it, and after washing the urea complex with water, coordinated urea bands are still observed at 1731 cm^−1^ (δsNH), 1656 cm^−1^ (δasNH), and 1595 cm^−1^ (νCO) (SI11). Therefore, these experiments indicate urea coordination to the nickel centers might not suffice to activate it to hydrolysis in which an outer-sphere encounter complex should be determined for these reactions. Kryatov et al. [41] stated that depending on the solvent, urea coordination could be complicated by solvation of the starting complex in which the higher ability to solvate the complex by a solvent resulted in slower ligand substitution reactions. This feature is expected as the stronger the bond between solvate and metal, the slower the formation of an inner-sphere complex by ligand substitution [42]. However, in the work of Kryatov et al., the authors could not detect the proposed intermediates due to a fast equilibrium and low yields of formation of these intermediates [41]. When analyzing the magnitude of the equilibrium constants, the low values (0.3 and 2.7 M^−1^, in acetone and in methanol, respectively) beyond indicating the formation of weak complexes between urea and 3d metals, could also suggest less stable intermediates, such as outer-sphere encounter complexe*s,* may be forming. Hence, we wonder if the outer-sphere activation of urea might be taking place in other urease mimics, which would indicate the major difference between urease enzyme and its mimics. (See Appendix A).

## 3. Materials and Methods

All reagents were from Sigma-Aldrich and were used without further purification, unless otherwise stated.

### 3.1. Synthesis of (1)

Triethylamine (Et*_3_*N, 100 μL, 0.7 mmol, 3.5 eq.) and 70 mg of nickel acetate (0.41 mmol, 2.1 eq.) were added to a solution containing 0.2 mmol of Trost (*RS*,*RS*)-Bis-ProPhenol Ligand (125 mg, 1 eq.) and 10 mL of acetonitrile (HPLC grade). The reaction proceeded under stirring for 24 h at room temperature. Then, cold diethyl ether was added to the reaction mixture to precipitate the impurities. After centrifugation, the organic layer was separated from the solid residue, dried with MgSO_4_, and filtered, and the solvent was removed until dryness, yielding 112 mg (58% yield) of the targeted complex. All attempts to isolate suitable crystals for X-ray analysis have not been successful so far.

C/H/N/Ni (calculated for [C_45_H_46_N_2_Ni_2_O_9_].3H_2_O.ACN): theor./exper.: 62.35/62.3, 6.21/5.98, 4.64/4.40 and 12.97/12.90.

IR: 3200–2800 cm^−1^ (C-H stretching), 1719 cm^−1^, 1671 cm^−1^, 1579 cm^−1^ (C = O asymmetric and symmetric stretching).

HRMS (m/z): 843.2178 (C_45_H_47_N_2_Ni_2_O_7_^+^ = 843.2090) and 797.2119 (C_44_H_45_N_2_Ni_2_O_5_^+^ = 797.2035). 

### 3.2. Synthesis of (2)

To a Schlenk flask, 97 mg of NiCl_2_.6H_2_O were added. The solid was heated to 50 °C for one hour under a high vacuum, converting it into a yellow solid. Anhydrous acetonitrile (10 mL) was then added to the flask, followed by addition of 0.2 mmol of Trost (*RS*,*RS*)-Bis-ProPhenol Ligand (125 mg, 1 eq.) and 28 μL of triethylamine (1 eq.). The reaction proceeded under stirring for 24 h at 40 °C. After this period, the solution was filtered, and the filtrate concentrated to 5 mL. Cold diethyl ether was then added to the reaction mixture. After filtration, any insoluble material was removed and the solvent evaporated to dryness, yielding 35 mg (36% yield) of the desired complex. All attempts to isolate suitable crystals for X-ray analysis have not been successful so far.

C/H/N (calculated for [C_55_H_74_ClN_4_Ni_2_O_3_].) theor./exper.:66.59/66.52; 7.52/7.69; 5.65/5.35

IR: 3063–2667 cm^−1^ (C-H), 2493 cm^−1^ (C = N)

LRMS (m/z): 468.22 (z = 2+) (C_51_H_70_N_3_Ni_2_O_6_), 671.22 (z = −1) (C_31_H_41_NNi_2_O_8_)

### 3.3. Synthesis of [Ni_2_L(OH)_2_]^−^ (5)

To 5 mL of a 0.1 mM solution of (1) in acetonitrile (conductivity: 40.2 μS cm^−1^), water aliquots were added (100 μL each) until reaching a stable conductivity of 132 μS cm^−1^. Any residual solid was filtered off, and the filtrate was dried for three days at high vacuum for a further reaction with urea.

### 3.4. Urea Hydrolysis with [Ni_2_L(OH)_2_]^−^ (5)

Complex (3) (2 mg, 2 μmol) was dissolved in 900 μL of anhydrous acetonitrile in a Schlenk flask under a nitrogen atmosphere. To this solution, 100 μL of a 60 mM urea solution (in anhydrous acetonitrile) was added, and the reaction mixture was kept at 20 °C for 10 minutes. After this period, a 100 μL aliquot of the reaction was analyzed for ammonia via the Berthelot method. Two control reactions were composed of: (a) 100 μL of a 60 mM urea solution (in anhydrous acetonitrile) and (b) addition of 100 μL of a 60 mM urea solution (in anhydrous acetonitrile) to a 900 μL solution of (1) (2 mM in anhydrous acetonitrile).

### 3.5. Determination of Urea Hydrolysis Using Bethelot Method

For the determination of ammonia formation, 100 μL of the complex dissolved in acetonitrile (2.8 mM) and 900 μL of urea solution (60 mM) were mixed in a 5 mL flask with stirring. The reaction was kept at 20 °C. Aliquots of 100 μL of the reaction were taken at different reaction times (10, 30, 60, 120, 120, 480, and 600 s) and were mixed with 250 μL of hypochlorite solution (2.5%) and 250 μL of sodium citrate/NaOH solution (0.38 M and 0.46 M). After mixing this solution, 300μL of sodium salicylate (2.75 M)/sodium nitroprussiate (9.4 μM) solution was added. After incubation of the solution at 20 °C for 15 minutes, the solution was analyzed by UV-Vis spectroscopy, monitoring the band absorption at 650 nm. Quantification was performed by means of a calibration curve with seven points (R^2^ = 0.99). Control reactions without the catalysts were also performed. All analyses were performed in triplicate.

### 3.6. General Protocol for Substrates Hydrolysis

A total of 50 mol of the substrate was added to a flask containing 1 mL of phosphate buffer solution (PBS, 20 mM, pH 7). After temperature stabilization at 25 °C, 100 L of the catalyst solution (urease (16 mM) or mimetic (2.2 mM)) was added and the reaction initiated. A 50 μL sample was taken from the reaction mixture after 10, 20, 30, 60, 120, and 180 s of reaction and analyzed by the indophenol method to determine the quantity of ammonia formed during the reaction. Amines or alcohols expected to form in the reaction were quantified by gas-chromatography analysis. All the hydrolysis reactions were performed in triplicate.

N-methyl urea, n-butylcarbamate, and acetamide hydrolysis: These reactions were also tested in 12 h of reaction at 25 °C and 50 °C.

N-phenyl urea, N-allylurea hydrolysis: These reactions were also tested in 12 h of reaction at 50 °C.

Benzyl benzylcarbamate and benzyl p-toluylcarbamate hydrolysis: These reactions were also tested at different pHs (3, 6, 9, and 12), temperatures (25, 35, and 45 °C), and solvents (ethanol, water, iso-propanol, and ethylacetate).

### 3.7. ^15^N-NMR Experiments

^15^N-urea (10 mg) was dissolved in 900 L of a 1:1 mixture of CD_3_CN/water and was analyzed in a 400 MHz Bruker spectrometer. After addition of 100 μL of a 1 mM acetonitrile solution of the complexes to the urea solution, the ^15^N NMR experiment was immediately reset. The ^15^N chemical shifts were referenced to the nitromethane.

### 3.8. Susceptibility Measurements

The dc magnetic susceptibility was measured over 2.0–300 K with a superconducting quantum interference device magnetometer (Quantum Design MPMS).

### 3.9. Evaluation of Substrate Hydrolysis by GC-MS

After the reaction from 4.6 was finished, it was analyzed by GC analysis using an achiral capillary column. The GC-MS conditions were carrier gas-H_2_, 100 kPa, injector 220 °C, detector 220 °C. A method composed of an isotherm at 180 °C was employed.

### 3.10. Computational—Geometry Optimization and Transition States

Calculations of full geometry optimization with the Density Functional Theory (DFT), using B3LYP exchange-correlation functional [43], and SVP function basis sets were performed for the [Ni2L] alone and associated with acetate, urea, and water. Their respective transition states (TS) were obtained and compared with reagents and products. All calculations were carried out using the TURBOMOLE 4.3 [44,45] and Gaussian 9.0 (mainly for TS refinement) [46] program packages as considered by us previously [47,48].

## 4. Conclusions

Two new dinuclear nickel complexes were synthesized and characterized. Both complexes were able to produce ammonia from urea solution. We have observed ligand substitution from acetate to chloride had a deep impact in the inorganic urease models, enabling a faster reaction rate with chloride as the more labile ligand. Both complexes were shown to produce isocyanate in the ^15^N experiments, but the fast ligand exchange in (2) complex resulted in a stationary equilibrium between urea and products, which could be easily shifted by pH decrease or bicarbonate addition. The equilibrium shift indicated isocyanate could suffer hydrolysis in the reaction conditions. When N-phenylurea was employed as a substrate, phenyl isocyanide and aniline could be detected by gas chromatography, corroborating the isocyanate hydrolysis hypothesis. Isocyanate hydrolysis by a urease analogue was first described in this work, revealing the geometry between coordinated water and urea is possibly different from the ones obtained in urease enzyme. Therefore, urea hydrolysis performed by complexes (1) and (2) seems to involve a different mechanism than the one from urease enzyme. Further refinements in ligand secondary coordination sphere could have an influence to mimic urease enzyme.

## Figures and Tables

**Figure 1 molecules-28-01659-f001:**
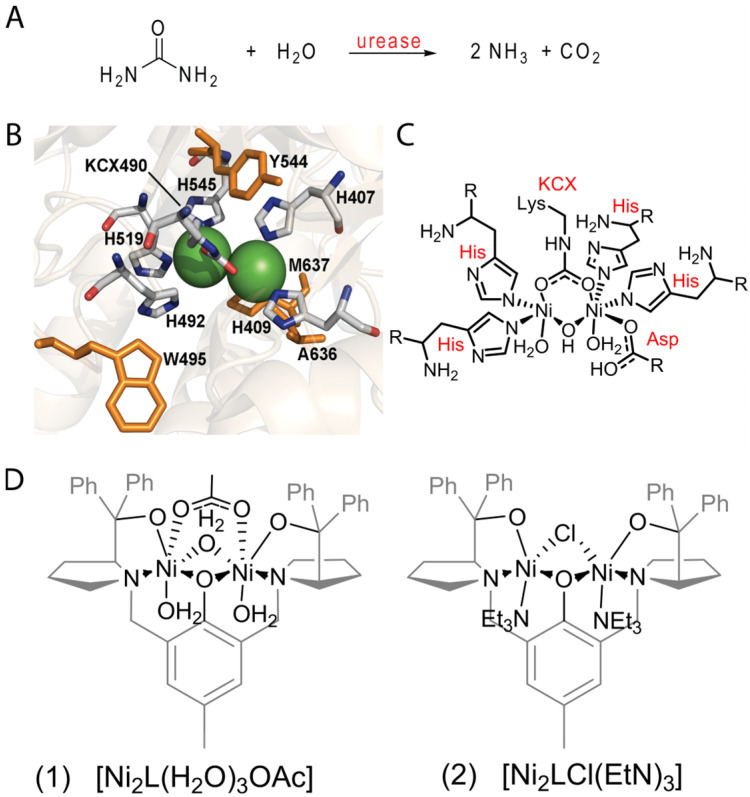
Scheme of the reaction performed by urease enzyme (**A**), generic active site of a urease enzyme. In orange are the hydrophobic amino acids which surround the active site (**B**,**C**) and *bis*-nickel complexes, [Ni_2_L(H_2_O)_3_(OAc)] (1) and [Ni_2_L(Cl)(Et_3_N)_2_](2) (2), based on (*RS*,*RS*)-Trost-bis-ProPhenol (L) ligand (**D**).

**Figure 2 molecules-28-01659-f002:**
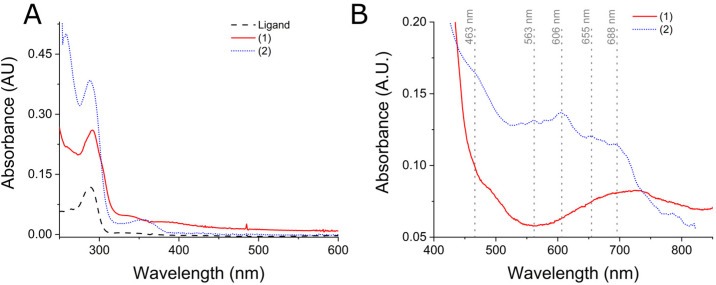
(**A**) UV-Vis spectra of the ligand (dashed black line), (1) (solid red line), and (2) (dotted blue line) in acetonitrile. (**B**) enlarged 450–850 cm^−1^ absorption region.

**Figure 3 molecules-28-01659-f003:**
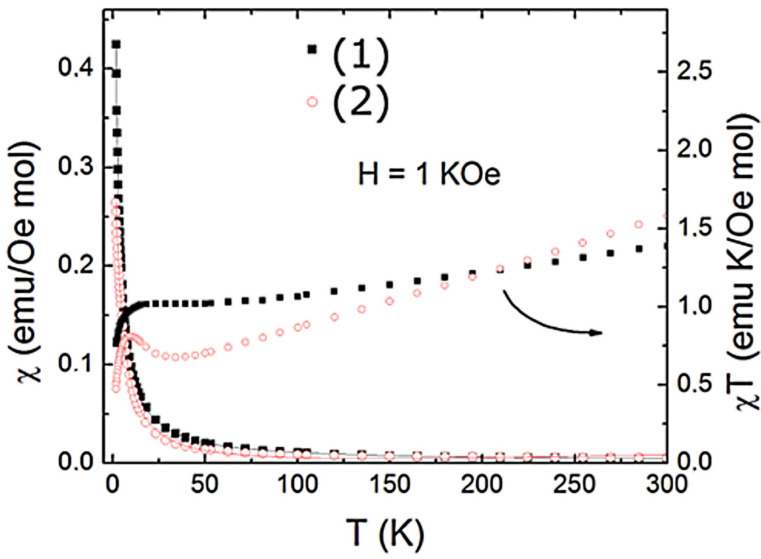
Temperature dependence of the magnetic susceptibility for (1) and (2). The solid lines represent the fits based on Equation (1).

**Figure 4 molecules-28-01659-f004:**
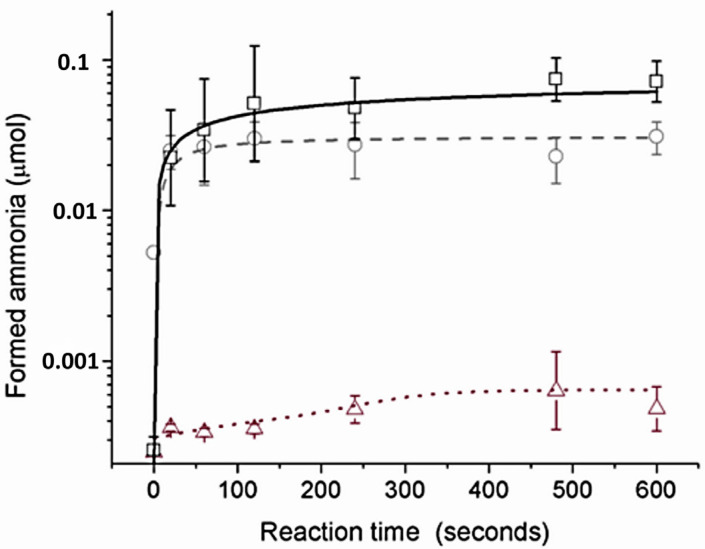
Quantification of formed ammonia over reaction time with urease enzyme (solid black line, □), with (2) (gray dashed line, ○), and with (1) (red dotted line, △). Error bars gave the mean of three independent catalytic experiments.

**Figure 5 molecules-28-01659-f005:**
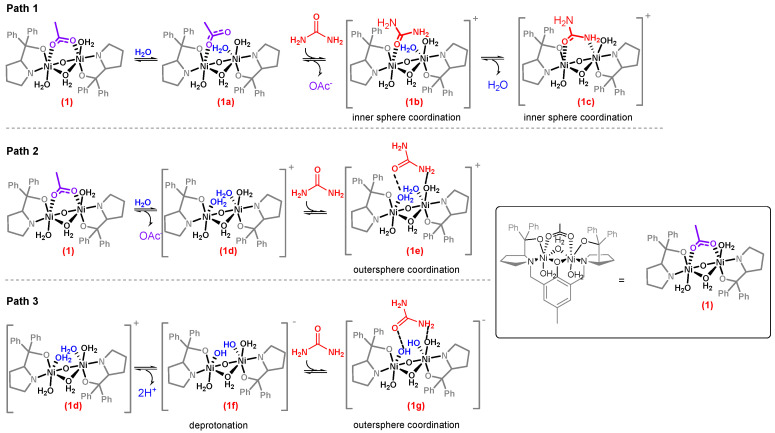
Possible intermediates formed during urea decomposition by (1) complex. Intermediate it 1c is formed upon water and urea coordination to the nickel center via intermediates 1b or 1d, whereas intermediates 1e and 1g are formed with secondary interactions between urea and coordinated water. Subfigure in the box shows the complete molecular representation and its equivalent form involved in catalysis.

**Figure 6 molecules-28-01659-f006:**
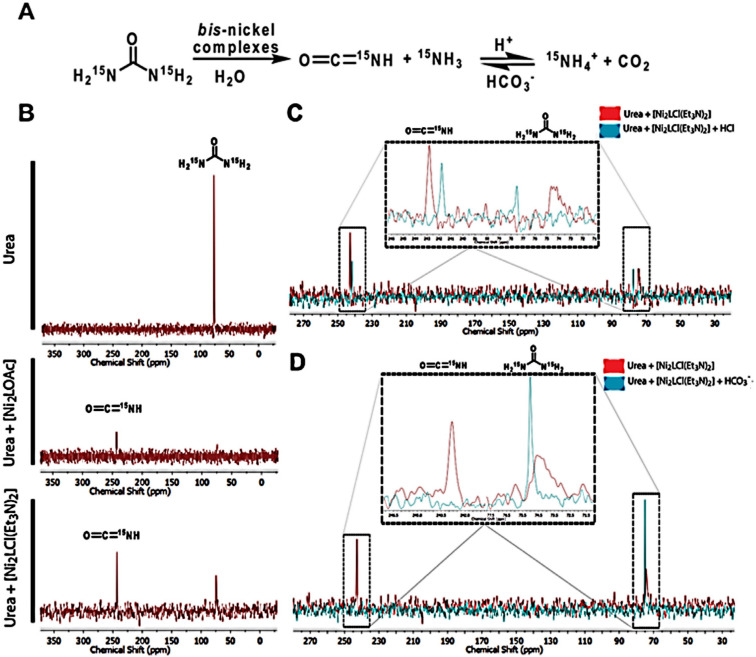
(**A**) Proposed reaction scheme of urea degradation by the bis-nickel complexes. (**B**) ^15^N NMR spectra of urea before and after the addition of complexes (1) and (2). (**C**) ^15^N NMR spectra of the reaction between urea and (2) complex before and after the addition of acid and (**D**) ^15^N NMR spectra of the reaction between urea and (2) complex before and after the addition of bicarbonate to shift the equilibrium.

**Figure 7 molecules-28-01659-f007:**
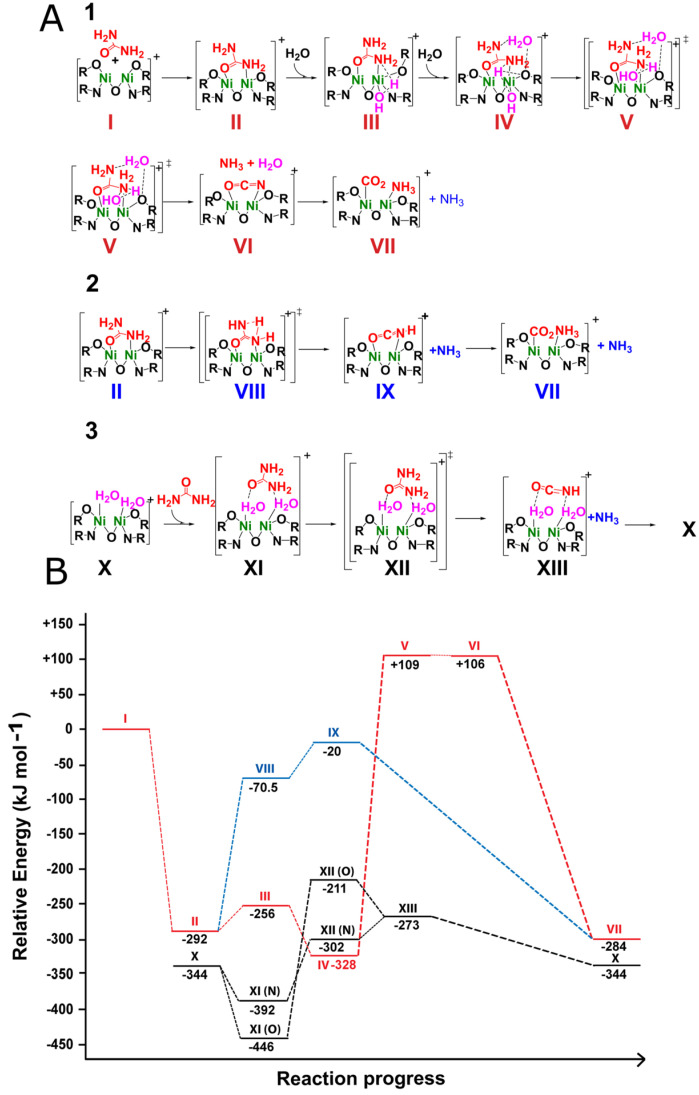
Three possible mechanism pathways (**A**) and their corresponding reaction energy diagrams: (1) urea coordination followed by water coordination to promote the elimination mechanism forming isocyanate, which is further hydrolyzed into carbon dioxide and ammonia. (2) Urea-coordinated self-elimination mechanism. (3) Water coordination followed by urea interaction, through oxygen or nitrogen to undergo an elimination mechanism forming isocyanate. (**B**) Theoretical energies calculated for the several reaction pathways.

**Table 1 molecules-28-01659-t001:** Hydrolysis of several substrates by urease (1.6 mM) and the bis-nickel complexes (2.2 mM) at different reaction times.

		(1)		(2)		Urease	
				Reaction Yield (%)		(±2)	
	Time	5 min	12 h	5 min	12 h	5 min	12 h
Substrate							
Urea		67	64	58	48	92	99
Formamide		70	66	62	87	73	21
Acetamide		6	4	0	4	12	82
N-methylurea		4	6	0	6	6	9
N-phenylurea ^c^		13 ^a^	13.7 ^a^	13.2 ^a^	14 ^a^	0	0
		8.6 ^b^	10.6 ^b^	9.9 ^b^	9.6 ^b^		
Butilcarbamate		0	0	0	0	0	0

^a^ Phenyl isocyanate; ^b^ aniline; ^c^ products were detected by GC-MS.

## Data Availability

The data presented in this study are available in this manuscript and in the support information.

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
