# Peer review of "Urea Decomposition Mechanism by Dinuclear Nickel Complexes"

_molecules, 2023, doi:10.3390/molecules28041659_

Round 1

Reviewer 1 Report

The paper entitled “Urea decomposition mechanism by dinuclear nickel complexes” by Christian O. Martin et al reports the preparation of two new compounds, [Ni2L(OAc)] and [Ni2L(Cl)(Et3N)2], where L= Trost bis-Pro-Phenol ligand, and their study as urease models to enlighten about urea decomposition mechanism.

The data analysis indicates that the lability of the ancillary substituents has a major role in the reaction kinetics and that the urea hydrolysis mechanism observed in Ni2L(OAc)] and [Ni2L(Cl)(Et3N)2] is possibly different from urease hydrolysis mechanism.

Perhaps the greatest contribution to the issue herein studied is the awareness of the importance of the secondary coordination sphere in the design of urease models. In this sense I find the work of general importance for the scientific community and meritorious to be published in “molecules, after a minor correction, mainly some typos.

Author Response

Reviewer 1

Perhaps the greatest contribution to the issue herein studied is the awareness of the importance of the secondary coordination sphere in the design of urease models. In this sense I find the work of general importance for the scientific community and meritorious to be published in “molecules, after a minor correction, mainly some typos.

Thank you for your comments. We revised the manuscript for typos and sent it to a native English speaker for the correction of language errors.

Reviewer 2 Report

This paper by Netto, Toma and coworkers deals with the nickel catalysed hydrolysis of urea from a biomimetic point of view, sheding light on several mechanistic aspects of the reaction with reference to urease. 

Although I find that the huge amount of experimental data provided by the authors, which encompasses almost all characterisation techniques, is admirable, I find the discussion to long and somehow confused.

My recommendation is to reorganise the manuscript, to join and resume the "results" and "discussion" sections, and correct errors such as: "neutral solution"; "despite some of ... were able"; "ancillary ligand" (this term is normally used for "ligands that provide an appropriate electronic and steric environment around the central metal atom or to help stabilize the central atom during reactions", not for Cl- or acetate"); etc.

I have not understood the connection between low symmetry and the Uv-vis spectra of the complexes.

In the absence of X ray characterisation I would avoid direct comparisons with other structures.

Experimental part should also be revised: NEt3 100 L; (di)ethyl ether added to remove impurities (I suppose it is use to precicpitate the product).

"neutral solution" (line 81).

In summary, I would be glad to provide a review of a revised and more concise version of the manuscript, which is not suitable of publication in Molecules as it is.

English, should be also thoroughly revised.

Author Response

Reviewer 2

My recommendation is to reorganise the manuscript, to join and resume the "results" and "discussion" sections, and correct errors such as: "neutral solution"; "despite some of ... were able"; "ancillary ligand" (this term is normally used for "ligands that provide an appropriate electronic and steric environment around the central metal atom or to help stabilize the central atom during reactions", not for Cl- or acetate"); etc.

Thank you for your comments. We believe they allowed the improvement of the manuscript and joined the “results” and “discussion” sessions. We also revised the terminology and corrected them.

I have not understood the connection between low symmetry and the Uv-vis spectra of the complexes.

We are sorry for the confusion. We wanted to state that complex [Ni2L(OAc)] has two symmetric nickel centers, whereas [Ni2L(Cl)(Et3N)2] has two asymmetric nickel centers. We expanded the discussion of the UV-Vis spectroscopy a little bit in an attempt to avoid this confusion.

In the absence of X ray characterisation I would avoid direct comparisons with other structures.

Thank you for your comment. We agree with the reviewer and removed the structural comparisons of the manuscript.

Experimental part should also be revised: NEt3 100 L; (di)ethyl ether added to remove impurities (I suppose it is use to precicpitate the product).

"neutral solution" (line 81).

The typo “L” and not “mL” is confusing to us, as in our version of the manuscript we have it as “mL”. We revised the whole manuscript and corrected the suggested phrases and words. Moreover, we sent it to a native English speaker for the correction of language errors.

Reviewer 3 Report

The ms. ‘’Urea decomposition mechanism by dinuclear nickel complexes’’ by Christian O. Martin et.al. describes the synthesis and characterization of two new Ni(II) complexes derived from reactions between Ni(II) species and the ligand Trost bis-Pro-Phenol ligand (L). The authors also tried to shed light into the urea decomposition mechanism of these two complexes. Both mimetic complexes produced ammonia from urea, in which the [Ni2L(OAc)] and [Ni2L(Cl)(Et3N)2] were ten and four times slower than urease, respectively. They observed also, that ligand substitution from acetate to chloride had a deep impact in the inorganic urease models, enabling a faster reaction rate with chloride as the more labile ligand. Both complexes were shown to produce isocyanate in the 15N experiments, but the fast ligand exchange in [Ni2L(Cl)(Et3N)2] complex resulted in a stationary equilibrium between urea and products, which could be easily shifted by pH decrease or bicarbonate addition. Therefore, urea hydrolysis performed by complexes [Ni2L(OAc)] and [Ni2L(Cl)(Et3N)2] seems to involve a different mechanism than the one from urease enzyme, and further refinements in ligand secondary coordination sphere could have an influence to mimic urease enzyme.

This work is well written and all the parts are described adequate. All the conclusions are accompanied by experimental data and prove all the findings. On the other hand, all the data are based on the assumption that these two complexes have the types [Ni2L(OAc)] and [Ni2L(Cl)(Et3N)2], with no available x-ray diffraction data. The formation of these two complexes has been confirmed with various techniques, but clearly the formation of 2 is questioned. The author claim that in MeCN solution complex 2 undergoes ligand substitution of Cl ligand with MeCN solvent. How sure are the authors that there is no H2O ligand in the structure? Also, in materials and methods section there are several typos with L and μL and the reference list should be revised. Moreover, in Figure 1 D the type of complex is [Ni2L(Cl)(Et3N)] instead of [Ni2L(Cl)(EtN)]. Also, in line 184 and 185 ‘’bands at 3432 and 3335 cm-1 for free NH2 , 3210 cm-1’’ are not shown in Fig. SI7.

Author Response

Reviewer 3

On the other hand, all the data are based on the assumption that these two complexes have the types [Ni2L(OAc)] and [Ni2L(Cl)(Et3N)2], with no available x-ray diffraction data. The formation of these two complexes has been confirmed with various techniques, but clearly the formation of 2 is questioned.

Thank you for your comments. We agree that a crystal structure of the complexes would improve the characterization of the complexes. In fact, we attempted the crystallization of the complexes under different conditions of solvents and temperatures and were unable to crystallize it. Complexes of bis-ProPhenol ligand are difficult to crystallize and to our knowledge there is only one X—ray structure of such complexes described at doi:10.1002/chem.200401159. However, even under the conditions of this paper, the complexes did not crystallize. Therefore, the structural characterization of our complexes is based on the FTIR, microanalysis, UV-Vis, magnetic susceptibility, electrochemistry, spectroelectrochemistry, conductivity, and mass spectroscopy, confirming their formation.

The author claim that in MeCN solution complex 2 undergoes ligand substitution of Cl ligand with MeCN solvent. How sure are the authors that there is no H2O ligand in the structure?

We agree with the reviewer that any ligand that can substitute the chloride might take its position. In fact, we did not use anhydrous acetonitrile in some of these experiments and chloride could be substituted by water molecules. For that reason, we write that chloride is substituted by solvent molecules, indicating that any coordinating solvent could have a similar result: charged solution and removal of chloride from the coordinating sphere.

Also, in materials and methods section there are several typos with L and μL and the reference list should be revised. Moreover, in Figure 1 D the type of complex is [Ni2L(Cl)(Et3N)] instead of [Ni2L(Cl)(EtN)]. Also, in line 184 and 185 ‘’bands at 3432 and 3335 cm-1 for free NH2 , 3210 cm-1’’ are not shown in Fig. SI7.

Thank you for calling our attention to these mistakes. We revised the manuscript accordingly, adding the missing figure, correcting Figure 1D and replacing typos.

Round 2

Reviewer 2 Report

20: from urea, being [...] and [...]....

33: 5-coordinate

72: help to

77: in the presence of ...., respectively.

81: neutral (instead of uncharged)

83: did not undergo dissociation into ionic species (instead of produced a solution...)

191: bonded or ligated

More comments:

- I do not unterstand why the two complexes (which would preferably indicated with numbers) are formulated without H2O molecules. Analyeses reveal that at least the acetate one cointains 3 of them.

- In figure 1 C, the first complex is represented with 3 terminal H2O ligand. In view of its symmetric properties, why is not the third H20 representes as a bridging ligand? Moreover, both representations should be improved, enlarged maybe, and the stereo-bonds correctly used.

- avoid expressions like "neutral solution", it sounds as referring to pH

- in figure 2 it is difficult to recognise the absorptions listed in the text

- prefer "UV-vis spectra" to "electronic spectra"

- line 177: I did not understand the statement as the conclusion of the results described before: "thus indicating...". 

- Figure 5: why is the OAc ligand represented as terminal in 1? In general I find this scheme quite confusing. Again, stereochemistry of the species represented here is not clear.

- Figure 7: Improve the ChemDraw representation, they look too small. I believe that, in the intermediate XI, the correct formula of urea is +NH3-CO-CH, or a H+ should be formed.

-line 323: what is the startig complex? V?

-In general, I think that the description of the mechanism should be improved with clear references to the calculated species. For example, is the energy difference of 52 kJ/mol  between 5 plus urea and XI?

-line 388: the molar ratio Trost ligand to NEt3 is 1:1. The complex contains 2 amines per ligand, though.

Barrios is cited a couple of times. Lippard should me mentioned instead, who is the corresponding author of the paper and one of the most recognised bio-inorganic chemists ever.

Author Response

Comments and Suggestions for Authors

20: from urea, being [...] and [...]....

33: 5-coordinate

72: help to

77: in the presence of ...., respectively.

81: neutral (instead of uncharged)

83: did not undergo dissociation into ionic species (instead of produced a solution...)

191: bonded or ligated

Thank you for pointing these mistakes. We corrected them all.

More comments:

- I do not unterstand why the two complexes (which would preferably indicated with numbers) are formulated without H2O molecules. Analyeses reveal that at least the acetate one cointains 3 of them.

Thank you, we corrected the formulas.

- In figure 1 C, the first complex is represented with 3 terminal H2O ligand. In view of its symmetric properties, why is not the third H20 representes as a bridging ligand? Moreover, both representations should be improved, enlarged maybe, and the stereo-bonds correctly used.

Thank you for your suggestion. We corrected the figure with the stereochemistry of the ligand as shown by Trost and Bartlett (dx.doi.org/10.1021/ar500374r | Acc. Chem. Res. 2015, 48, 688−701). We understand your comment about the bridged water and decided to represent it as a bridged ligand, but without a crystal structure it is hard to know the correct representation of the coordination of the third water molecule.

- avoid expressions like "neutral solution", it sounds as referring to pH

Thank you. We corrected this expression.

- in figure 2 it is difficult to recognise the absorptions listed in the text

Thank you. We changed figure 2 to improve the visualization of the bands.

- prefer "UV-vis spectra" to "electronic spectra"

We changed the terminology.

- line 177: I did not understand the statement as the conclusion of the results described before: "thus indicating...". 

We changed this phrase.

- Figure 5: why is the OAc ligand represented as terminal in 1? In general I find this scheme quite confusing. Again, stereochemistry of the species represented here is not clear.

Considering your comment, we are now representing the complex through a different view in the hope of making it clearer. We also redrew the scheme.

- Figure 7: Improve the ChemDraw representation, they look too small. I believe that, in the intermediate XI, the correct formula of urea is +NH3-CO-CH, or a H+ should be formed.

We enlarged the representations and corrected the intermediate formulations.

-line 323: what is the startig complex? V?

We added the number of the states in the text to clarify the discussion.

-In general, I think that the description of the mechanism should be improved with clear references to the calculated species. For example, is the energy difference of 52 kJ/mol  between 5 plus urea and XI?

We added the number of the states in the text to clarify the discussion.

-line 388: the molar ratio Trost ligand to NEt3 is 1:1. The complex contains 2 amines per ligand, though.

Indeed the complex has two triethylammines per complex, which would indicate a maximum yield of 50% of the complex by using the stoichiometry 1:1. As described in section 3.2, our yield is 36%, in agreement to that.

Barrios is cited a couple of times. Lippard should me mentioned instead, who is the corresponding author of the paper and one of the most recognised bio-inorganic chemists ever.

We added Lippard name to the referencing too.

Reviewer 3 Report

The ms ''Urea decomposition mechanism by dinuclear nickel complexes'' has been significantly improved since the last time it was submitted. The authors managed to respond in all the reviewer's comments and thus I believe it can be now accepted for publication in its present form. I suggest the authors to perform a carefull check in english terminology used in the main text of this ms and typos that are scattered in text, as well as the reference list.

Author Response

Comments and Suggestions for Authors

The ms ''Urea decomposition mechanism by dinuclear nickel complexes'' has been significantly improved since the last time it was submitted. The authors managed to respond in all the reviewer's comments and thus I believe it can be now accepted for publication in its present form. I suggest the authors to perform a carefull check in english terminology used in the main text of this ms and typos that are scattered in text, as well as the reference list.

Thank you. We performed a carefull check in english terminology, as suggested.